# Rural Homecare Nurses’ Challenges in Providing Seamless Patient Care in Rural Japan

**DOI:** 10.3390/ijerph17249330

**Published:** 2020-12-13

**Authors:** Ryuichi Ohta, Yoshinori Ryu, Takuji Katsube, Chiaki Sano

**Affiliations:** 1Community Care, Unnan City Hospital, 96-1 Iida, Daito-cho, Unnan, Shimane Prefecture 699-1221, Japan; yoshiyoshiryuryu.hpydys@gmail.com (Y.R.); katsube-takuji@city.unnan.shimane.jp (T.K.); 2Department of Community Medicine Management, Faculty of Medicine, Shimane University, 89-1 Enya cho, Izumo, Shimane Prefecture 690-0823, Japan; sanochi@med.shimane-u.ac.jp

**Keywords:** interprofessional collaboration, homecare nurses, multimorbidity, patient care, quality of care, rural medical care

## Abstract

Homecare nurses manage patients with extreme homecare dependence through interprofessional collaboration. The quality of the collaboration depends on situations, and the difficulties of homecare nurses are complicated in rural settings because of a few healthcare resources. This study determined rural homecare nurses’ difficulties during interprofessional collaboration in providing seamless patient care. Focus groups, followed by one-on-one interviews, were conducted with 13 rural homecare nurses working in rural Japan. Using thematic analysis, four themes were extracted: collaboration with physicians, the collaboration with the government, the collaboration with care workers, and the collaboration among hospital nurses. Rural homecare nurses have difficulties in their working relationships with other professionals, with vague definitions of each professional’s roles and responsibilities, and with information-sharing. Interprofessional education and information-sharing should respect rural professional and cultural backgrounds. Respect can accomplish mutual understanding among professional care, leading to seamless patient care in rural home care.

## 1. Introduction

Homecare nurses manage homecare patients who have conditions that make them highly dependent on a nurse’s care. With the aging of the population, the number of older patients is notably increasing [1]. In Japan, which is one of the most aging countries, the need for home care has increased. In home care, homecare patients can use various services from healthcare professionals, such as physicians, nurses, home care workers, care managers, and therapists. The services are mostly covered by social insurance and long-term care insurances supported by the Japanese government [2]. These patients have various medical problems, called multimorbidity, which make homecare professionals’ work complicated [3,4]. In hospital care, physicians can lead a medical team, followed by other healthcare professionals. However, in home care, patients have various problems regarding not only multimorbidity but also psychosocial problems, which cannot be dealt with by physician-led medical teams [4]. Homecare nurses have medical and social knowledge of diseases related to homecare [4,5]. They can thus deal with patients’ psychosocial problems in patients’ homes through the communication with various healthcare professionals. Their comprehensive approaches to patients’ biopsychosocial problems may improve patient care, but the burden on them has become more substantial than that in the past [6,7]. To reduce this burden, interprofessional collaboration with other medical and care professionals is essential [8].

In home patient care, effective interprofessional collaboration drives the quality of care [9,10]. Effective collaboration with physicians improves patients’ medical care by approaching the frequently changing conditions of patients with multimorbidity [9,11]. To address typical biopsychosocial problems efficiently, homecare workers and care managers are essential, because they can transfer home care patients to long-term facilities for day services and rehabilitation and convey information about subtle patient changes to homecare nurses and solve psychosocial problems using social resources. Additionally, care managers do care coordination in home care [12,13,14]. To collaborate with each professional efficiently, homecare nurses must understand each professional’s skills and settings depending on the situation. In home care, compared with acute care in hospitals or institutionalized long-term care, healthcare professionals may act independently in patients’ home, and homecare nurses have to discuss patients’ medical conditions with various professionals belonging to other organizations [12,13,14]. Their communication should be performed efficiently and effectively, which can make homecare nurses’ interprofessional collaboration specific. However, there are few standards of formal information sharing in Japanese home care. 

Interprofessional collaboration can be affected by contextual factors, such as location, population, and medical resources, which may make rural medical conditions specific [12]. In rural areas, the rate of aging may be higher than that in urban areas, and medical resources are limited, which makes homecare challenging [14,15]. These factors can affect collaboration among various types of medical professionals; by respecting each professional’s capacities during interprofessional collaboration, professionals can compensate for each other’s limitations and improve the quality and seamlessness of patient care [16,17]. Previous studies investigated the difficulties and challenges in rural interprofessional collaboration in Japan, such as physicians, home care workers, and care managers [12,13,18]. However, there is a lack of evidence about the present conditions of interprofessional collaboration in rural areas, especially regarding homecare nurses’ difficulties in providing seamless patient care. Clarifying the difficulties facing rural homecare nurses could contribute to effective discussions regarding future home care in rural settings. Further clarification of this issue may help facilitate the provision of the seamless home care needed to address the problems various countries face in creating sustainable aging societies. This study aimed to examine rural homecare nurses’ difficulties in providing seamless patient care during interprofessional collaboration.

## 2. Materials and Methods

To examine homecare nurses’ difficulties in interprofessional collaboration, 2 focus groups and 10 one-on-one semi-structured interviews were conducted with homecare nurses in a rural area in Japan. Focus groups and one-on-one interviews were 40 and 30 minutes, respectively. This study used purposive sampling. The participants were recruited from among homecare nurses who worked in Unnan City. The Unnan nursing homecare section sent invitation letters to each homecare nursing station. At the time of this study, there were 15 homecare nurses in Unnan City. The participants voluntarily participated in the focus group discussion. After the focus groups, the participants were invited to participate in one-on-one interviews, to which 10 of them agreed. 

### 2.1. Setting

Unnan is in the eastern part of Shimane Prefecture, on the southern border of Hiroshima Prefecture. Its total land area is 553.1 km^2^, accounting for 8.3% of Shimane Prefecture, most of which is under forest cover. A survey conducted in 2017 revealed that the total population of Unnan was 38,882 (18,720 male and 20,162 female), and the aging rate was 37.82%. Unnan City Hospital is the only general public hospital in the city. Other facilities in the city include 16 clinics, 3 visiting nurse stations, and 12 homecare stations.

### 2.2. Focus Groups

The focus group theme was rural homecare nurses’ difficulties in interprofessional collaboration. One participant facilitator and a participant who took down the minutes were chosen randomly in each group. Before each focus group, the researchers provided four questions for the facilitators to lead the group: “With which professionals do you experience difficulties in collaborating”; “When do you experience the difficulties”; “How do you experience the difficulties”; and “Why do you think that there are such difficulties in your interprofessional collaboration?” Each group engaged in open-ended discussions about their difficulties for 40 minutes.

### 2.3. One-on-one Semi-structured Interviews

Based on the results of the focus groups, interviews were conducted to determine the participants’ concrete experiences regarding each theme and concept. In interprofessional collaboration, relationships among same professionals can affect their discussion, and each professional can have emotional and individual issues to other professionals, which they do not want to express in the front of their professionals. So, new concepts and themes were investigated in one-on-one interviews [12,13]. The interviewees were recruited from the focus groups. Theoretical sampling was used to select the participants based on their age, sex, and duration of work experience. In these interviews, after the results of the focus groups were presented to the interviewees, they were asked about their actual experiences and ideas and other difficulties with interprofessional collaboration 

### 2.4. Analysis

Thematic analysis was used to extract themes and concepts related to homecare nurses’ perceptions about interprofessional collaboration. The analysis consisted of the following phases: familiarization with the data, generation of initial codes, searching for themes, reviewing themes, defining and naming categories, and producing the report, including a selection of illustrative data and quotations [19]. The content of the focus groups and one-on-one interviews was used in this analysis. All content was transcribed verbatim. First, the content from the two focus groups was analyzed to familiarize the authors with and enhance their knowledge about the homecare nurses’ ideas regarding interprofessional collaboration difficulties. Two authors independently coded the transcripts, and then checked for agreement on their open coding. Thereafter, they discussed the open codes and emerging concepts and categorizations, identified themes, and reached a consensus by recoding or redefining concepts and themes they disagreed about the coding. This process was repeated for the content of each focus group. Subsequently, one-on-one interviews were conducted to further inquire about homecare nurses’ difficulties using the same process employed in the focus groups. The interviews were repeated until no new concepts or themes appeared. For member checking, the analysis was provided to the interviewees, whose feedback was included in the final revision of the themes and concepts. Eventually, no new themes emerged during the member checking phase. Finally, the themes and concepts were discussed and agreed upon by all the authors.

### 2.5. Ethical Considerations

Participants were informed in advance that all collected data would be used only for research purposes. The study’s aims, data usage, and methods of protecting personal information were explained before participants provided written consent. The Unnan City Hospital Clinical Ethics Committee approved this study (approval number 20170024).

## 3. Results

Among the 15 homecare nurses in Unnan, 13 participated in the focus group discussions. Twelve participants were female, and the average age was 45.4 years old (standard deviation [SD] = 11.5). The average duration of nursing experience was 22.8 years (SD = 11.0), and that of homecare nursing experience was 6.3 years (SD = 6.5). Two focus group interviews were conducted, one with six participants and the other with seven. Ten participants were interviewed individually. The average age was 43.7 years old (SD = 12.3). The average duration of nursing experience was 21.7 years (SD = 12.2), and that of homecare nursing experience was 6.8 years (SD = 5.1) (Table 1). Including repeated interviews, 17 interviews were performed in total. Based on the thematic analysis, 4 themes and 10 concepts emerged (Table 2). 

### 3.1. Collaboration with Physicians

#### 3.1.1. Different Standards

Homecare nurses and physicians are specialists in medical issues, and can share medical information easily. However, the homecare nurses felt that there were some difficulties in conveying clinical changes in patients to physicians because each rural physician had different needs regarding patient information. Several participants stated: 

Each rural physician wants to get patients’ clinical information based on their own interests. To manage homecare patients, some common points should be shared, but some physicians need information about patients’ conditions different from that which homecare nurses want to share with them, or they do not listen to the information we give them. (Focus Group 1)

Young doctors realize the importance of interprofessional collaboration. However, in rural settings, there are many old type physicians who do not know interprofessional collaboration because of the lack of official learning. The lack of learning can inhibit the effective collaboration. (Participant E, one-on-one interview)

Rural homecare nurses and physicians collaborated, but in various situations, they had different standards for assessing patients, which could be dangerous for patients. Additionally, they realized the lack of the education of interprofessional collaboration.

#### 3.1.2. Professional Hierarchy

Initially, physicians decide patients’ treatments, and homecare nurses follow their instructions. To effectively care for homecare patients, homecare nurses and physicians should share information and decide on patients’ care by coordinating with the patients and their families. Patients tend to respect physicians’ opinions but not homecare nurses’ advice, which can lead to weak collaboration between physicians and nurses. Several participants stated: 

I think, homecare nurses visit patients’ homes more frequently than physicians and can quickly detect subtle changes that should be addressed by medical professionals. However, even if I give advice, some patients do not follow [it] and wait for physicians’ examinations before doing anything. (Focus Group 2)

Patients cannot help obeying physicians’ suggestions based on their culture. Even if I suggest some advice for their health, patients tend to persist on physicians’ advice. When there is the gap between physicians and nurses in patient care, effective care may not be performed because of the hierarchy. (Participant G, one-on-one interview)

Although strong relationships between patients and physicians contribute to the continuity of care, swift action to treat changing symptoms cannot be taken.

### 3.2. Collaboration with Government

#### 3.2.1. Disruption of Information-sharing Due to Frequent Job Changes

Effective collaboration between local government and homecare nurses is essential for the best use of nursing resources. In usual conditions, local governments financially support homecare nurses to sustain the ongoing management of homecare nursing stations. Local governments provide rural citizens with information regarding homecare and how to use it. Local governments support dependent patients in their home by quickly connecting them with related departments and providing financial and social support. Homecare nursing stations make the work appealing inside and outside their regions to recruit new nurses. However, in rural homecare, the situations were different. Several participants stated:

In local government, clerks’ jobs change frequently. Even if we discuss future projects for conciseness, we sometimes cannot proceed with them effectively because of disruption information between former and present government employees. (Focus Group 1)

Especially, in taking care of difficult patients with chronic neurological diseases, who needs much support, local government officer’s actions are very slow. These patients need comprehensive care from help of various departments of city hall. The delay can be from unsmooth collaboration in local governments (Participant E, one-on-one interview).

Disruptions in the flow of information with local government prevent homecare workers from managing their nursing stations by employing a constant workforce to accommodate those who need intensive home care. Additionally, ineffective collaboration in local governments impinged on dependent patients’ care in their home.

#### 3.2.2. Gaps Between Departments in Local Government

Information-sharing occurs not only between homecare nurses and local government workers but also within local government. However, as each worker’s job is divided into tasks across different departments, there are various stakeholders, which hinders collaboration. One participant stated:

Even if one department in the local government is motivated to change something, substantial discussion is needed to decide on one thing. Although this is a common thing in local government, it is challenging for me to decide on our plans for collaboration with them. (Focus Group 1)

Homecare nurses and local government have different ideas regarding organization and the decision-making process. Vague decision-making processes in local government in relation to outsiders have thus impeded collaboration.

### 3.3. Collaboration with Care Professionals

#### 3.3.1. Different Criteria Regarding the Judgment of Patients’ Conditions

Homecare nurses acquire information from care professionals, such as homecare workers and care managers, about subtle changes in their patients’ medical conditions. Their collaboration can lead to the early detection of acute conditions and quick medical treatment. However, there are differences in criteria between homecare nurses and care professionals that lead to challenges in treating homecare patients quickly by identifying subtle symptoms. One participant stated: 

Home care nursing is vital for homecare patients, as home care nurses go to their homes more frequently than other professionals. However, I cannot share proper information, partially because of the differences in professional backgrounds. I judge the patients’ conditions by our criteria, which are not shared. The differences sometimes cause deterioration of the condition of homecare patients, especially when patients cannot say their symptoms clearly. (Participant B, one-on-one interview)

The gap in medical knowledge and perceptions about homecare patients leads to misunderstandings about the patients, which causes difficulties in maintaining the acute sensitive conditions of homecare patients.

#### 3.3.2. Vague Role Differences Between Nurses and Care Workers

Typically, medical issues are approached by homecare nurses; however, there are some situations in which homecare nurses and homecare workers can do the same things. As the Japanese healthcare system is gradually changing, the criteria for each professional’s limitations are vague. One participant stated:

Regarding the medical issues of homecare patients, there are common things that both homecare nurses and workers can do. If I understand each other’s range of work, I can collaborate more effectively. However, home care nurses do not know each other’s settings and working conditions well. (Participant F, one-on-one interview)

A clear view of each professional’s competency could drive their collaboration, as they would not need to engage in redundant work. However, the situation does not currently allow for these groups to sufficiently understand each other. Both types of professionals provide the same kinds of care to their patients, which is unnecessarily time-consuming for homecare nurses and workers alike. 

#### 3.3.3. Inconsistency in the Content of shared information

Information-sharing between homecare nurses and workers is vital, but the content varies depending on the homecare workers and situations because of little standard of information sharing between them. Information sharing was different in each home care team. This lack of consistent content can lead to misunderstandings related to patients’ conditions. Several participants stated:

I think, the consistency of the information is important between home care nurse and homecare workers. Sometimes, important patient information is not conveyed, and trivial information is frequently conveyed. This may confuse homecare nurses when they make decisions. (Participant A, one-on-one interview)

The information sharing system has been advanced. Some of rural healthcare professionals use information and communication technology (ICT) to share patients’ information each other. However, I feel, the way of thinking about conditions regarding patients can be different among healthcare professionals. Lack of understanding each other can be critical for interprofessional collaboration. (Focus Group 2)

Gathering and sharing information requires a mutual understanding of one another’s working conditions and needs for specific information. Although ICT can be useful, the effective usage of the ICT needs the mutual understanding among healthcare professionals. Sharing redundant information—and not sharing essential information—regarding critical patients can place patients at risk and induce frustration for nurses.

### 3.4. Collaboration with Hospital Nurses

#### 3.4.1. Differences in Understanding Home Care

Nurses learn basic knowledge about home care in nursing school; therefore, they should also understand what home care is like by observing homecare nurses as part of their education. However, after becoming nurses, hospital nurses have few experiences with home care and are unable to imagine how patients are taken care of in their own homes. One participant stated:

Various types of care are possible in homecare settings, similar to hospital care. Thanks to home care, hospitalized patients can get the opportunity to go home. However, I think, because of hospital nurses’ lack of knowledge about the realities of home care, they may inhibit their patient from going back to their home. I consider that it can be also caused by their busy working conditions and their careers as hospital nurses (Participant D, one-on-one interview).

Hospital nurses’ limited consideration of the possibilities for applying home care with their patients results in fewer patients receiving home care and less collaboration between homecare and hospital nurses. Hospital nurses’ working conditions are so challenging that they do not have many opportunities to consider their patients’ homecare conditions, and they primarily focus on patients’ medical conditions.

#### 3.4.2. Lack of Opportunities to Share Information

Homecare nurses work in patients’ homes, and hospital nurses work in hospitals. Thus, there are few interactions between them or opportunities to understand each other’s jobs and working conditions. As their work experience grows, the discrepancy between the two settings widens. One participant stated:

To understand each other, nurses should cyclically learn both home care and hospital care. Even if it may not be possible, home care nurses should have more opportunities to share each other’s conditions to understand each other. Now, those opportunities are few. I think, as nurses’ experience grows, they tend not to understand each other (Participant G, one-on-one interview).

The lack of nurses in rural areas makes them busy, leading to difficulties in interactions and collaborations among them, which widens their distance. For efficient collaboration, dialogue between home care and hospital nurses is needed, which could allow for more hospitalized patients to go home, based on their wishes.

## 4. Discussion

This study shows that rural home care nurses experience difficulties in interprofessional collaboration with physicians, home care workers, care managers, hospital nurses, and local government. For the collaboration with physicians, hierarchy and difference in patients’ information needs appeared. For the collaboration with the government, inadequate information continuity and gaps between departments appeared. For the collaboration with care managers and care workers, difference in criteria, vague role differences, and inconsistent information-sharing appeared. For the collaboration among hospital nurses, difference in understanding home care, and insufficient opportunities for sharing information appeared. For better collaboration, overcoming the hierarchy between professionals, understanding one another’s professions in depth, and an effective information-sharing system are needed.

This study shows that hierarchy exists between rural physicians and nurses in patients’ care. Professional hierarchy is a common feature in interprofessional collaboration, especially concerning medical and care professionals in rural settings, especially Japan [18,20,21]. This hierarchy can be increased when in dealing with patients’ medical problems, because care professionals may not necessarily be competent in treating some medical issues [20]. As the participant stated, the hierarchy between physicians and nurses is traditionally strong in the medical field, based on the structure of the medical profession, especially in rural Japanese areas. As physicians authorize all patients’ treatments, and nurses follow their orders [22]. In present Japanese medical education, interprofessional education has been encouraged, which can change the present interprofessional collaboration in Japan. In rural settings like this study, rural older physicians may not learn in depth interprofessional collaboration, which may inhibit rural interprofessional collaboration [18]. Although this professional hierarchy may be inevitable, the clarification of individual professional roles is one solution to the related conflicts and responsibilities through interprofessional collaboration [23,24]. In modern medicine, elderly patients with multimorbidity have complicated problems that multiple professionals deal with together [25]. To improve rural interprofessional collaboration, educational interventions among rural healthcare staff should be performed for the clarification of individual professional roles. This finding emphasized the importance of clearly defined roles for all professionals in rural settings, which can enhance their overall effectiveness. Future research can inquire into ways to improve each professional’s focus on their own role through implementing sessions of interprofessional education among them, which can lead to fewer problems regarding responsibilities and conflicts.

Creating a place where each working condition can be shared personally is essential. Each region has its own specific medical and care resources, which directly affect medical and care professionals’ working conditions, particularly in rural areas [26,27]. In addition to clarifying each professional’s responsibilities and roles, their work limitations and efficiency can be discussed based on specific homecare cases. In-person case-based discussions between medical and care professionals can facilitate the sharing their working conditions and creating unique approaches to home care [17,28]. Through these discussions, they could grow to know each other in depth, which may lead to a mutual understanding of one another, not only as professionals but also as people [29]. Thus, the local governments should officially enhance their discussion, and regular interprofessional discussions should be conducted for sharing each other’s knowledge, skills, and difficulties to improve interprofessional collaboration in rural settings [30]. 

Effective information-sharing can optimize interprofessional collaboration. To share patient information effectively, the information should fit each medical and care professional’s needs [31]. Based on this study, there may be some discrepancies between the needs of medical and care professionals in rural settings. Additionally, regarding the usage of ICT, as this study’s participants stated, rural interprofessional collaboration can be driven by ICT with mutual understanding among rural health professionals. Therefore, what each professional should share through interprofessional collaboration should be discussed during regular interprofessional discussions [18]. The use of ICT could lead to quick and versatile information-sharing among professionals [29,32]. As information-sharing methods may depend on medical and care professionals’ personal characteristics, such as age and educational background, the particular methods can be discussed in each setting [33]. Regarding the usage of ICT, the hierarchy among rural healthcare professionals is critical and impinges on the effective usage of ICT [33]. In the rural setting of the present study, ICT is used for collaboration among healthcare professionals. Using ICT with a mutual understanding among rural professionals can be vital for rural home care [34,35,36]. For the effective usage of ICT, the establishment of effective relationship on the regular basis among them important. Based on the relationship, the way of ICT usage should be discussed. Governments should apply an official information sharing system, such as ICT, by facilitating regular meetings among healthcare professionals [30]. Thus, future studies could examine the effectiveness of ICT-driven interprofessional collaboration in rural settings.

This study has several limitations. First, its participants were a selection of homecare nurses from one rural Japanese city. This could lead to the extraction of specific ideas about interprofessional collaboration. Data were collected mostly from homecare nurses in their workplaces, which could lead to one comprehensive dataset about rural homecare nursing. Second, we formed two focus groups, which might not lead to the saturation of concepts. This study used two methods to obtain data from the participants for the methodological triangulation. For the saturation, the following study should obtain more participants as interviewees up to 12 based on the previous article [37]. This study also examined the present conditions of interprofessional collaboration between homecare nurses through one-on-one semi-structured interviews, and the solutions for better collaboration. Further research could inquire about other aspects of homecare nursing regarding interprofessional collaboration and solutions for effective interprofessional collaboration.

## 5. Conclusions

This study shows that rural homecare nurses have difficulties in their working relationships with other medical and care professionals, with vague definitions of each professional’s roles and responsibilities, and with information-sharing. To address these problems, a mutual understanding of other professionals and consistent information-sharing systems are needed. Moreover, for interprofessional collaboration, constant interprofessional education among rural healthcare professionals should be performed. The contents of information-sharing through the ICT should be discussed daily among rural healthcare professionals. Through effective interprofessional education and information-sharing, mutual understanding and seamless patient care can be accomplished in rural home care.

## Figures and Tables

**Table 1 ijerph-17-09330-t001:** The demographic of the participants.

Variable	Focus Group	One-on-one Interview
Age, years, mean (SD)	45.4 (11.5)	43.7 (12.3)
Clinical experience, years, mean (SD)	22.8 (11.0)	21.7 (12.2)
homecare nursing experience, years, mean (SD)	6.3 (6.5)	6.8 (5.1)

**Table 2 ijerph-17-09330-t002:** Results of thematic analysis.

Theme	Concepts
Collaboration with physicians	Different standards
	Professional hierarchy
	One-way information-sharing
Collaboration with government	Disruption of information-sharing due to frequent job changes
	The gap between departments in city hall
Collaboration with care professionals	Different criteria regarding the judgement of patients’ conditions
	Vague role differences between nurses and care workers
	Inconsistency in the content of shared information
Collaboration with hospital nurses	Difference in understanding about home care
	Lack of opportunities to share information

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
