# Peer review of "Rural Homecare Nurses’ Challenges in Providing Seamless Patient Care in Rural Japan"

_ijerph, 2020, doi:10.3390/ijerph17249330_

Round 1
Reviewer 1 Report
Thank you for an overall interesting topic with relevance for health care research and family medicine.
Background section: While reading the results, I became painfully aware of my lack of understanding of how the Japanese health care system works and how home care is organized. This was especially true when you talked about local governmental strucutres. Could you provide a very short overview of how home care works in your local enivonment in the background section for somebody that is not familiar with it? This would make understanding the results much more concrete, should help your discussion and provide an interesting comparison with other countries (I am German; And I can relate to the problems you stated, it's quite similar here).
L. 72: Include a complete focus group questionnaire (lead questions) if available as an appendix
L. 80-86 and L 99-100: How were the different sources triangulated (were interview partners asked about aspects of the focus groups / topics that emerged in the analysis)? Please be more precise. I would assume that focus groups were used to review how topics were negotiated and how consensus was build between participants, while the interviews focused more on narrative experiences. It would be desirable to emphasize the contribution of each data collection method to the study. For example, my exception would be that the experiences of difficulties in home care stem mainly from the interviews, while arguments and negotiations about possible solutions would stem from the focus groups. The results section has mainly interview excerpts and often clearly state that statements were made by "one participant" - to my mind, you miss an opportunity to introduce whenever consensus was reached in the focus groups or a broader range of participants agreed or disagreed on a topic (which is a specialty of focus groups) - since you transcribed the data verbatim you should have no problem revisiting your text excerpts and put the into context.
Had it been preferable to make an additional focus group after the interviews , present the results and talk about solutions (member check)? Why was that left out? Please discuss that as a limitation.
L. 108: I would propose stating participants of each data collection method independently (give a result table for focus group participants and the interview participants). This should later be discussed in limitations.
L.231-234 It is unclear if this statement is based on the results or general considerations. Consider rephrasing the first paragraph as a one-sentence summary of the results.
L. 235-248 This paragraph is quite vague and reads as a background section. It illustrates a lack of focus. Please consider quoting some of the literature in the background section (maybe with relation to home care in japanese rural. settings) Your focus is a rural home care setting in a specific Japanese urbal environment - that is not a weakness, but a strength of your article and quite agreeable in qualitative research. Please discuss the topics with this focus in mind: What is specific for Japan? What is specific for Japanese Home Care? How have other researchers approached the topic methodologically and content-wise? What are the new contribution of your research to this specific settings?
L. 266 here you introduce ICT as a concept, however this has not been named before, neither in the results nor in the background section. Was it a topic of your study aim? Put it in the background section. Was it a result of the interviews/focus groups? Introduce it as a result. Otherwise, please reconsider.
L 271- I would expect a discussion of sample size, triangulation of data sources and collection methods. Refer to "guided by information power" bei K Malterud
In summary, I would like to se a greater emphasis on the local environment and the problem of rural home care in the context of the Japanese health care system with its specifics. You might even want to explore cultural aspects in greater detail. I would like to encourage you to rework several sections of the manuscript.
Author Response
Thank you for an overall interesting topic with relevance for health care research and family medicine.
Background section: While reading the results, I became painfully aware of my lack of understanding of how the Japanese health care system works and how home care is organized. This was especially true when you talked about local governmental strucutres. Could you provide a very short overview of how home care works in your local enivonment in the background section for somebody that is not familiar with it? This would make understanding the results much more concrete, should help your discussion and provide an interesting comparison with other countries (I am German; And I can relate to the problems you stated, it's quite similar here).
Response:
We thank the reviewer for this insightful comment. We agree with the suggestion. In response to this comment, we have added to the background section, in lines 31 to 35, information regarding the provision of home care under the Japanese health care system.
- 72: Include a complete focus group questionnaire (lead questions) if available as an appendix
Response:
We thank the reviewer for this insightful comment. We agree with the suggestion. In response to this comment, we have added all of the questions that were used in the focus group in lines 96 to 99.
- 80-86 and L 99-100: How were the different sources triangulated (were interview partners asked about aspects of the focus groups / topics that emerged in the analysis)? Please be more precise. I would assume that focus groups were used to review how topics were negotiated and how consensus was build between participants, while the interviews focused more on narrative experiences. It would be desirable to emphasize the contribution of each data collection method to the study. For example, my exception would be that the experiences of difficulties in home care stem mainly from the interviews, while arguments and negotiations about possible solutions would stem from the focus groups. The results section has mainly interview excerpts and often clearly state that statements were made by "one participant" - to my mind, you miss an opportunity to introduce whenever consensus was reached in the focus groups or a broader range of participants agreed or disagreed on a topic (which is a specialty of focus groups) - since you transcribed the data verbatim you should have no problem revisiting your text excerpts and put the into context.
Response:
We thank the reviewer for this insightful comment. We agree with the suggestion. In response to this comment, we have added the reasons to use both focus group and one-on-one interviews in lines 102 to 107.
Had it been preferable to make an additional focus group after the interviews, present the results and talk about solutions (member check)? Why was that left out? Please discuss that as a limitation.
Response:
We thank the reviewer for this insightful comment. We agree with the suggestion. In response to this comment, we have added to the discussion section in lines 349 to 357, the description of the solution as a limitation and an issue to be addressed in the future research.
- 108: I would propose stating participants of each data collection method independently (give a result table for focus group participants and the interview participants). This should later be discussed in limitations.
Response:
We thank the reviewer for this insightful comment. We agree with the suggestion. In response to this comment, we have summarized the demographics of the participants in lines 134 to 139, as well as in Table 1.
L.231-234 It is unclear if this statement is based on the results or general considerations. Consider rephrasing the first paragraph as a one-sentence summary of the results.
Response:
We thank the reviewer for this insightful comment. We agree with the suggestion. In response to this comment, we have revised the description in the first paragraph of the discussion section by summarizing the results in lines 286 to 296.
- 235-248 This paragraph is quite vague and reads as a background section. It illustrates a lack of focus. Please consider quoting some of the literature in the background section (maybe with relation to home care in japanese rural. settings) Your focus is a rural home care setting in a specific Japanese urbal environment - that is not a weakness, but a strength of your article and quite agreeable in qualitative research. Please discuss the topics with this focus in mind: What is specific for Japan? What is specific for Japanese Home Care? How have other researchers approached the topic methodologically and content-wise? What are the new contribution of your research to this specific settings?
Response:
We thank the reviewer for this insightful comment. We agree with the suggestion. In response to this comment, we have revised the description in the second paragraph of the discussion section by adding specific features of our research, in lines 297 to 318.
- 266 here you introduce ICT as a concept, however this has not been named before, neither in the results nor in the background section. Was it a topic of your study aim? Put it in the background section. Was it a result of the interviews/focus groups? Introduce it as a result. Otherwise, please reconsider.
Response:
We thank the reviewer for this insightful comment. We agree with the suggestion. In response to this comment, we have added a description of ICT in the discussion and result sections, in lines 333-335 and 342-345.
L 271- I would expect a discussion of sample size, triangulation of data sources and collection methods. Refer to "guided by information power" bei K Malterud
Response:
We thank the reviewer for this insightful comment. We agree with the suggestion. In response to this comment, we have added the description of sample size and methodological triangulation in lines 349 to 351.
In summary, I would like to see a greater emphasis on the local environment and the problem of rural home care in the context of the Japanese health care system with its specifics. You might even want to explore cultural aspects in greater detail. I would like to encourage you to rework several sections of the manuscript.
Response:
We thank the reviewer for this insightful comment. We agree with all of the suggestions and have revised our manuscript accordingly.
Reviewer 2 Report
The introduction section could be expanded to include a section specific to rural health challenges or rural health practice. Similarly, the discussion section is rather limited and should be expanded further.
Author Response
The introduction section could be expanded to include a section specific to rural health challenges or rural health practice. Similarly, the discussion section is rather limited and should be expanded further.
Response:
We thank the reviewer for this insightful comment. We agree with the suggestion. In response to this comment, we have revised the description in the introduction section by including rural health conditions and rural Japanese health care. In addition, we have revised the description in the discussion section by adding present and future rural interprofessional collaboration issues.
Reviewer 3 Report
This study would like to emphasize the importance of cross-profession collaborations in home care, especially in the rural area. The issue is important for quality of life, but there are some points needed to be clarified. Here are some comments and suggestions for the authors.
Introduction
- Although the issue is important for quality of care of home care, the literature review does not indicate the critical point and lack of theoretical basis. More discussion about the interprofessional collaboration for home care (especially in rural area) need to be addressed. For example, the medial care team is often lead by the physician from the viewpoint of medical sociology, but in home care, the home care nurse’s role is often more important. In the rural area, the resource is not abundant and more often the service coordination depends on care manager. The working pattern of acute care or institutionalized long-term care can be used for comparison with home care, and then the unique collaboration of home care can be highlighted.
- The background of Japan’s National Health Insurance or National Long-term Care Insurance about home care is suggested to be introduced in the Introduction. The incentives or difficulties in home care can be discussed.
Methods
- Please provide the reason of collecting data by both focus group and one-on-one interviews. Why did you use two qualitative methods in one study? What were the relationships of these two-stages of study procedures? If the cross-professional collaboration is the core issue of this study, why only the home care nurses were interviewed (but not the physicians and care workers)?
- Regarding one-on-one semi-structured interviews, the questions are suggested to be listed.
Results
- Section 3.2: Please state the usual procedure of home care under the government’s administration, so readers have a better understanding about the report from the participants.
- Section 3.3 & 3.4: Under the National Health Insurance or Long-Term Care Insurance system, there should be some standard procedure about the information recording. If there is a standard information system to share information under the government’s management and hospitals, please also provide the background. If not, regarding the 3.3.3 the shared information issue, was that only happened in one home care station or was common in all the home care stations?
- The study was conducted in a rural area. The special issue for rural area in providing home care should be analysed too. It seems the home care stations only have chance to collaborate with government and the only hospital. Did the home care stations also need to transfer the home care users to long-term care institutions?
- Were homecare nurses also the care managers, so they also were responsible for care coordination? Or there were the care managers in the provision of home care other than the homecare nurses?
Discussion
- I agree that ICT is a good way to solve the information sharing issue. Were there any homecare nurses reporting that they tried some ICT way to share the information for care coordination? The platform across different agencies can also be an issue for informational sharing.
- The authors may consider to add policy suggestions for the government or the health care system.
Author Response
This study would like to emphasize the importance of cross-profession collaborations in home care, especially in the rural area. The issue is important for quality of life, but there are some points needed to be clarified. Here are some comments and suggestions for the authors.
Introduction
- Although the issue is important for quality of care of home care, the literature review does not indicate the critical point and lack of theoretical basis. More discussion about the interprofessional collaboration for home care (especially in rural area) need to be addressed. For example, the medial care team is often lead by the physician from the viewpoint of medical sociology, but in home care, the home care nurse’s role is often more important. In the rural area, the resource is not abundant and more often the service coordination depends on care manager. The working pattern of acute care or institutionalized long-term care can be used for comparison with home care, and then the unique collaboration of home care can be highlighted.
Response:
We thank the reviewer for this insightful comment. We agree with the suggestion. In response to this comment, we have added to the introduction section a description about interprofessional collaboration and home care with respect to the specific conditions of rural medicine and homecare nurses.
- The background of Japan’s National Health Insurance or National Long-term Care Insurance about home care is suggested to be introduced in the Introduction. The incentives or difficulties in home care can be discussed.
Response:
We thank the reviewer for this insightful comment. We agree with the suggestion. In response to this comment, we have added to the background section, descriptions of Japan’s National Health Insurance or National Long-term Care Insurance, and the challenges of rural home care
Methods
- Please provide the reason of collecting data by both focus group and one-on-one interviews. Why did you use two qualitative methods in one study? What were the relationships of these two-stages of study procedures? If the cross-professional collaboration is the core issue of this study, why only the home care nurses were interviewed (but not the physicians and care workers)?
Response:
We thank the reviewer for this insightful comment. We agree with the suggestion. In response to this comment, we have added the reasons to use both focus group and one-on-one interviews in lines 102 to 107. In addition, we have added the reasons to focus on rural home care nurses to the background in lines 54 to 57.
- Regarding one-on-one semi-structured interviews, the questions are suggested to be listed.
Response:
We thank the reviewer for this insightful comment. We agree with the suggestion. In response to this comment, we have added all the questions that were used for the focus group in lines 96 to 99
Results
- Section 3.2: Please state the usual procedure of home care under the government’s administration, so readers have a better understanding about the report from the participants.
Response:
We thank the reviewer for this insightful comment. We agree with the suggestion. In response to this comment, we have added a description of the usual procedure of home care under the government’s administration, under Section 3.2.
- Section 3.3 & 3.4: Under the National Health Insurance or Long-Term Care Insurance system, there should be some standard procedure about the information recording. If there is a standard information system to share information under the government’s management and hospitals, please also provide the background. If not, regarding the 3.3.3 the shared information issue, was that only happened in one home care station or was common in all the home care stations?
Response:
We thank the reviewer for this insightful comment. We agree with the suggestion. In response to this comment, we have added the descriptions of a standard information system in Japanese home care and current conditions to the background and under Section 3.2, respectively.
- The study was conducted in a rural area. The special issue for rural area in providing home care should be analysed too. It seems the home care stations only have chance to collaborate with government and the only hospital. Did the home care stations also need to transfer the home care users to long-term care institutions?
Response:
We thank the reviewer for this insightful comment. We agree with the suggestion. In Japanese contexts, transmission of home care patients is performed by home care workers or care managers. We have added a description of home care nurses’ work in Japanese home care to the background section.
- Were homecare nurses also the care managers, so they also were responsible for care coordination? Or there were the care managers in the provision of home care other than the homecare nurses?
Response:
We thank the reviewer for this insightful comment. We agree with the suggestion. In Japanese contexts, care managers are responsible for care coordination in home care. We have added a description of care managers’ work in Japanese home care to the background section.
Discussion
- I agree that ICT is a good way to solve the information sharing issue. Were there any homecare nurses reporting that they tried some ICT way to share the information for care coordination? The platform across different agencies can also be an issue for informational sharing.
Response:
We thank the reviewer for this insightful comment. We agree with the suggestion. In response to this comment, we have added the description of ICT in the discussion and result parts, in lines 333-335 and 342-345.
- The authors may consider to add policy suggestions for the government or the health care system.
Response:
We thank the reviewer for this insightful comment. We agree with the suggestion. In response to this comment, we have added a description of policy suggestions for the government or the health care system, to the discussion section.
Round 2
Reviewer 1 Report
Thank you for this revision, the article makes more sense now and one can now generally get a better idea of what the setting is and how the design works.
The background section is better, but somewhat redundant in some aspects and the paragraphs are lengthy. I suggest shortening it and removing sentences that are not absolutely essential to the study and the aim.
There still are quite a few language corrections to be made. I am no native speaker myself, so I am very aware of the difficulties, but please ask a native speaker for a plain english correction , it will improve the readability and help gain a greater audience for your interesting contribution.
The methods section has improved greatly!
L.246-248: This excerpt does not read like a verbatim transcript of a focus group, more like a summary (in fact, most of the statements in the results section do).
I would recommend exclusively using excerpts of the verbatim text in your results section (check LL.165-168 and compare with the excerpt L-246-248 - notice the difference? "I" and "we" as opposed to a kind of summary.
Otherwise, if you did in fact summarize the verbatim texts for the results section please state in the methods section how you summarized them (right now: L. 118: "All content was transcribed verbatim").
In any case, I would strongly recommend that all results should be presented in the same manner and using the same style. My preference is verbatim text, because it puts the reader in the focus group. If there are ethical and/or content concerns that warrant a summary, please describe the operationalization in the methods section. If you changed the verbatim text for the results section, that is - to my mind- a limination that needs to be discussed: Who summarized? Did you re-check the summaries with the focus group participants and interview partners (member check)? How did you proceed if any discrepancies arose?
L329-344: COnsindering the focus on digital technology and information sharing, I believe this is a point you want to make. However, I believe that your results more show that digitalisation is not the problem or the soluttion, but that interprofessional cooperation needs to start with the people or participants: If we have bad interprofessional cooperation and hierarchy, digital information exchange simply does not help. If you digitalize bad communication, you have digital bad communication. I do believe that is a relevant point to be made and is supported by your results, but you should present it accordingly. Try to wrap this section around those thoughts if you agree and see if you can find any interesting research on that topic (maybe you find something that disagrees with that notion?)
358-366 the same notion as stated in the above paragraph should also be in the conclusion
Lastly, upon re-reading your manuscript and now re-reading the abstract, I believe you must adapt it to the changes you made in the main text. I'll give point-by-point suggestions to that:
12 Abstract: Homecare nurses manage patients with extreme homecare dependence. Effective
13 interprofessional collaboration between doctors and homecare workers drives care quality. This
- what about Japan and the special setting there that you described in the background?
14 study determined rural homecare nurses’ difficulties during interprofessional collaboration in
Suggestion: Formulate this as the study aim
15 providing seamless patient care. Focus groups and one-on-one interviews were conducted with
16 rural homecare nurses working in rural Japan. Using thematic analysis, four themes were extracted.
Please try to work in how you triangulated the focus groups and interviews (study design). I would also suggest putting in the number of participants.
17 For the collaboration with physicians, hierarchy and difference in patients’ information needs
18 appeared. For the collaboration with the government, inadequate information continuity and gaps
19 between departments appeared. For the collaboration with care workers, difference in criteria,
20 vague role differences, and inconsistent information-sharing appeared. For the collaboration
21 among hospital nurses, difference in understanding home care, and insufficient opportunities for
22 sharing information appeared. Addressing these issues necessitates a deep understanding of each
23 profession and the development of consistent information-sharing systems, which would lead to
24 seamless patient care.
YOu should try to shorten this results summary and put in th eparticipants' perspective (these issues did not magically 'appear', they were stated by your participants. Maybe just enumerate the themes, but make clear these are based on the results.
Last, after shortening the results section of the abstract, add a few sentences of your conclusions so that people with little time can skim your abstract and get a basic understanding of your contribution to the scientific discourse.
My suggestion is a minor revision, reducing redundancies and adressing the issues in the results section plus a thorough language check by a native speaker. Keep at it, good work!
Author Response
Reviewer 1
Thank you for this revision, the article makes more sense now, and one can now generally get a better idea of what the setting is and how the design works.
The background section is better but somewhat redundant in some aspects, and the paragraphs are lengthy. I suggest shortening it and removing sentences that are not absolutely essential to the study and the aim.
Response:
We thank the reviewer for this insightful comment. We agree with the suggestion. In response to this comment, we have shortened the background section by deleting redundant parts.
There still are quite a few language corrections to be made. I am no native speaker myself, so I am very aware of the difficulties, but please ask a native speaker for a plain English correction. It will improve the readability and help gain a greater audience for your interesting contribution.
Response:
We thank the reviewer for this insightful comment. We agree with the suggestion. In response to this comment, we have consulted the English editing company for a plain English correction.
The methods section has improved greatly!
L.246-248: This excerpt does not read like a verbatim transcript of a focus group, more like a summary (in fact, most of the statements in the results section do). I would recommend exclusively using excerpts of the verbatim text in your results section (check LL.165-168 and compare with the excerpt L-246-248 - notice the difference? "I" and "we" as opposed to a kind of summary. Otherwise, if you did, in fact, summarize the verbatim texts for the results section, please state in the Methods section how you summarized them (right now: L. 118: "All content was transcribed verbatim"). In any case, I would strongly recommend that all results should be presented in the same manner and using the same style. My preference is verbatim text because it puts the reader in the focus group. If there are ethical and/or content concerns that warrant a summary, please describe the methods section's operationalization. If you changed the verbatim text for the results section, that is - to my mind- a limitation that needs to be discussed: Who summarized? Did you re-check the summaries with the focus group participants and interview partners (member check)? How did you proceed if any discrepancies arose?
Response:
We thank the reviewer for this insightful comment. We agree with the suggestion. In response to this comment, we have reviewed all of the excerpts and revised the contents. Furthermore, we have added the issue regarding member checking in the analysis section.
L329-344: considering the focus on digital technology and information sharing, I believe this is a point you want to make. However, I believe that your results show that digitalization is not the problem or the solution, but that interprofessional cooperation needs to start with the people or participants: If we have bad interprofessional cooperation and hierarchy, digital information exchange does not help. If you digitalize bad communication, you have digital bad communication. I believe that is a relevant point to be made and is supported by your results, but you should present it accordingly. Try to wrap this section around those thoughts if you agree and see if you can find any interesting research on that topic (maybe you find something that disagrees with that notion?) 358-366 the same notion as stated in the above paragraph should also be in conclusion.
Response:
We thank the reviewer for this insightful comment. We agree with the suggestion. In response to this comment, we have revised the paragraph regarding ICT, focusing on the relationship between effective relationships among professionals and ICT usage. Furthermore, we added the same contents to the conclusion.
Lastly, upon re-reading your manuscript and now re-reading the abstract, I believe you must adapt it to the changes you made in the main text. I'll give point-by-point suggestions to that: This study determined rural homecare nurses’ difficulties during interprofessional collaboration in Abstract: Homecare nurses manage patients with extreme homecare dependence. Effective interprofessional collaboration between doctors and home care workers drives care quality.
- what about Japan and the special setting there that you described in the background?
Response:
We thank the reviewer for this insightful comment. We agree with the suggestion. In response to this comment, we have revised the contents of the abstract, including rural contexts.
Suggestion: Formulate this as the study aim at providing seamless patient care. Focus groups and one-on-one interviews were conducted with rural homecare nurses working in rural Japan. Using thematic analysis, four themes were extracted. Please try to work on how you triangulated the focus groups and interviews (study design). I would also suggest putting in the number of participants.
Response:
We thank the reviewer for this insightful comment. We agree with the suggestion. In response to this comment, we have revised the abstract's content by triangulating the methods of focus groups and interviews and putting in the number of participants.
For the collaboration with physicians, hierarchy, and difference in patients’ information need appeared. For the collaboration with the government, inadequate information continuity and gaps between departments appeared. For the collaboration with care workers, differences in criteria, vague role differences, and inconsistent information-sharing appeared. For the collaboration among hospital nurses, understanding home care, and insufficient opportunities for sharing information appeared. Addressing these issues necessitates a deep understanding of each profession and the development of consistent information-sharing systems, which would lead to seamless patient care. It would be best if you tried to shorten this results summary and put in the participants' perspective (these issues did not magically 'appear', your participants stated them. Maybe enumerate the themes, but make clear these are based on the results.
Response:
We thank the reviewer for this insightful comment. We agree with the suggestion. In response to this comment, we have revised the abstract's content by shortening the parts of the results.
Last, after shortening the abstract results section, add a few sentences of your conclusions so that people with little time can skim your abstract and get a basic understanding of your contribution to the scientific discourse.
Response:
We thank the reviewer for this insightful comment. We agree with the suggestion. In response to this comment, we have added our contribution in the last part of the abstract.
My suggestion is a minor revision, reducing redundancies and addressing the results section's issues, plus a thorough language check by a native speaker. Keep at it. Good work!
Reviewer 2 Report
The authors appear to have addressed the concerns I previously noted.
Author Response
Response:
We thank the reviewer for your review.
Reviewer 3 Report
The authors have substantially revised the manuscript according to the suggestions or comments. The current manuscript is acceptable.
Author Response

(The authors gave the same response as above.)
